# Effects of Chemotherapy Agents on Circulating Leukocyte Populations: Potential Implications for the Success of CAR-T Cell Therapies

**DOI:** 10.3390/cancers13092225

**Published:** 2021-05-06

**Authors:** Nga T. H. Truong, Tessa Gargett, Michael P. Brown, Lisa M. Ebert

**Affiliations:** 1Translational Oncology Laboratory, Centre for Cancer Biology, University of South Australia and SA Pathology, North Terrace, Adelaide, SA 5000, Australia; nga.truong@sa.gov.au (N.T.H.T.); Tessa.Gargett@sa.gov.au (T.G.); MichaelP.Brown@sa.gov.au (M.P.B.); 2Cancer Clinical Trials Unit, Royal Adelaide Hospital, Port Rd, Adelaide, SA 5000, Australia; 3Adelaide Medical School, University of Adelaide, North Terrace, Adelaide, SA 5000, Australia

**Keywords:** immunotherapy, combination therapy, adoptive cell therapy, CAR-T cell therapy, chemotherapy, cytotoxic CD8^+^ T cells, tumour microenvironment, immunosuppression

## Abstract

**Simple Summary:**

CAR-T cell therapy is a new approach to cancer treatment that is based on manipulating a patient’s own T cells such that they become able to seek and destroy cancer cells in a highly specific manner. This approach is showing remarkable efficacy in treating some types of blood cancers but so far has been much less effective against solid cancers. Here, we review the diverse effects of chemotherapy agents on circulating leukocyte populations and find that, despite some negative effects over the short term, chemotherapy can favourably modulate the immune systems of cancer patients over the longer term. Since blood is the starting material for CAR-T cell production, we propose that these effects could significantly influence the success of manufacturing, and anti-cancer activity, of CAR-T cells. Thus, if timed correctly, chemotherapy-induced changes to circulating immune cells could allow CAR-T cells to unleash more effective anti-tumour responses.

**Abstract:**

Adoptive T-cell therapy using autologous T cells genetically modified to express cancer-specific chimeric antigen receptors (CAR) has emerged as a novel approach for cancer treatment. CAR-T cell therapy has been approved in several major jurisdictions for treating refractory or relapsed cases of B-cell precursor acute lymphoblastic leukaemia and diffuse large B-cell lymphoma. However, in solid cancer patients, several clinical studies of CAR-T cell therapy have demonstrated minimal therapeutic effects, thus encouraging interest in better integrating CAR-T cells with other treatments such as conventional cytotoxic chemotherapy. Increasing evidence shows that not only do chemotherapy drugs have tumoricidal effects, but also significantly modulate the immune system. Here, we discuss immunomodulatory effects of chemotherapy drugs on circulating leukocyte populations, including their ability to enhance cytotoxic effects and preserve the frequency of CD8^+^ T cells and to deplete immunosuppressive populations including regulatory T cells and myeloid-derived suppressor cells. By modulating the abundance and phenotype of leukocytes in the blood (the ‘raw material’ for CAR-T cell manufacturing), we propose that prior chemotherapy could facilitate production of the most effective CAR-T cell products. Further research is required to directly test this concept and identify strategies for the optimal integration of CAR-T cell therapies with cytotoxic chemotherapy for solid cancers.

## 1. Introduction

Despite significant advances in treatment, cancer is often an incurable disease. Empiric or rational combinations of the different modalities of anti-cancer treatment including surgery, radiotherapy, chemotherapy and immunotherapy may be used to enhance the overall therapeutic benefit while minimising the adverse effects of overlapping modes of action [1]. Although chemotherapy and radiotherapy can destroy cancers as external agents, immunotherapy, which is emerging as the newest therapeutic approach, relies on a patient’s own immune system to recognise, attack and kill tumour cells [2,3,4]. The most prominent immunotherapeutic approaches include immune checkpoint inhibitor therapy and adoptive cell therapy, primarily using chimeric antigen receptor (CAR)-T cells [4,5].

CAR-T cell therapy is approved for relapsed/refractory (r/r) B-cell leukaemia and lymphoma and, in time, may bring significant benefits for patients with solid cancers as well [6]. Currently, however, cytotoxic chemotherapy is a mainstay of treatment for most cancer types. The development of new CAR-T cell therapies should therefore take into account the effects of these agents because new cell therapies will almost certainly be trialled initially in the setting of relapse after standard treatment. Of particular note, chemotherapy has long been known to induce an immediate numerical depletion of leukocytes, mainly neutrophils, and the risk of neutropenia is routinely mitigated by immediate post-chemotherapy administration of granulocyte-colony stimulating factor (G-CSF). However, there is also increasing evidence of longer-term immunomodulation following recovery from the initial effects, which could potentially be beneficial for the use of immunotherapies [7,8,9].

Understanding the effects of chemotherapy drugs on the immune system will be important for maximising clinical benefit from all forms of cancer immunotherapy. Indeed, the transient effects of cytotoxic chemotherapy are recognized in the US Food and Drug Administration (FDA) product monographs for the CD19-CAR-T cell therapies, tisagenlecleucel (Kymriah^TM^, Novartis, Basel, Switzerland) and axicabtagene ciloleucel (Yescarta^TM^, Kite Pharma, Los Angeles, CA, USA). In reviewing the clinical studies supporting FDA approval of tisagenlecleucel, it is stated that “Bridging chemotherapy between leukapheresis and LD chemotherapy was permitted to control disease burden. LD chemotherapy could be omitted if the white blood cell count was <1000 cells/μL.” Whereas for axicabtagene ciloleucel, the product monograph states that “Bridging chemotherapy between leukapheresis and lymphodepleting chemotherapy was not permitted”. Furthermore, it should be mentioned that a number of chemo-immunotherapy regimens are now US FDA-approved as standard first-line treatment for patients with locally advanced or metastatic lung cancer [10,11,12,13]. The doublet cytotoxic chemotherapy, which is employed with an immune checkpoint inhibitor in these chemo-immunotherapy regimens, comprises a platinum drug and one of several different cytotoxic drugs, which may include pemetrexed, paclitaxel, vinorelbine, etoposide or gemcitabine. Accordingly, the compelling clinical and scientific rationale for standard chemo-immunotherapy [14,15] may also apply to the consideration of how best cytotoxic chemotherapy might be integrated with CAR-T cell therapy.

In this review, we discuss how conventional chemotherapy can have substantial and long-lasting effects on the phenotype and frequency of many circulating leukocyte populations, including T cells. Surprisingly, many of these effects are likely to have an overall positive effect on immune function. Moreover, since blood leukocytes comprise the starting population for the manufacture of CAR-T cells, we propose that these chemotherapy-induced changes could influence the anti-cancer activity of the final CAR-T cell product considerably. We conclude by considering how this information could be used to optimise the timing of blood collection for CAR-T cell manufacture, ultimately resulting in an enhanced CAR-T cell product and improved patient outcomes.

## 2. Immune Evasion in Cancer

The immune system has a critical role in inhibiting and sculpting the development, growth and invasion of malignant tumours [16,17]. Nascent transformed cells express altered phenotypes and thus can be detected and eliminated by immune cells. However, tumour cells, which have evaded immune surveillance, can survive and carry the gene mutations and phenotypic changes that are transmitted to their progeny. Subsequently, these abnormal cell lineages grow uncontrollably to form clinically apparent malignant tumours, in part because of insufficient recognition and elimination by the immune system.

Immune recognition of cancer cells relies to a large extent on the detection of tumour neoantigens, which derive from DNA alterations accumulated during the malignant transformation of cells [18,19]. However, some tumour types have an inherently low mutation rate and thus are poorly immunogenic. For example, pancreatic carcinoma and sarcoma have very low somatic mutation rates and are largely non-responsive to anti-PD1 immunotherapy, which relies on a pre-existing anti-tumour immune response for efficacy [20]. However, during tumour evolution and under immune selective pressure from tumour-infiltrating T cells, even highly immunogenic tumours can reduce neoantigen expression by mechanisms including promoter hypermethylation of mutant alleles. This reveals the active operation of an immunoediting process, thus facilitating escape from immune surveillance [21,22].

Loss of tumour immunogenicity can also result from defects in the antigen processing and presenting machinery (APM) of cancer cells, thus preventing T-cell recognition [23]. For example, genetic and epigenetic alterations result in aberrant peptide-transporting molecules that reduce antigen processing efficiency and lead to the expression of “peptide-free” MHC molecules on the cancer-cell surface [23]. Another mechanism leading to defective APM is downregulated expression of MHC molecules, which further hides cancer cells from T-cell recognition [23,24]. Cancer cells can also have reduced cell-surface expression of Fas molecules compared to normal cells [25]. Therefore, cytotoxic lymphocytes such as NK cells and cytotoxic CD8^+^ T cells cannot induce cancer cell apoptosis by using the Fas-FasL pathway [26]. In addition, cancer cells have downregulated expression of CD80 and CD86 molecules, which are required for co-stimulation of both T cells and NK cells [27]. Ultimately, corruption of APM reduces the immunogenicity of tumour cells, thus enabling immune evasion by, and persistent survival of, tumour cells, leading to uncontrolled tumour growth [17].

Cancer cells also acquire active immunosuppressive properties that further reduce their immunogenicity and create a suppressive tumour microenvironment [17]. For example, breast cancer cells and metastatic melanoma cells can release IL-10 to inhibit dendritic cell functions and the generation of local immune responses, thus helping to promote tumour growth [28,29,30]. In addition, cancer cells have been shown to upregulate indoleamine 2, 3-dioxygenase (IDO), an enzyme that plays a role in overall immune suppression [31,32,33]. Another suppressive mechanism that tumour cells exploit is to upregulate surface expression of PD-L1 [34]. Binding of tumour-cell PD-L1 to the immune checkpoint molecule PD-1, which is expressed on antigen-specific T cells, inhibits T-cell activation and expansion, thus permitting unchecked tumour growth [35,36]. Tumour cells also exploit other well-characterised immune checkpoint molecules such as CTLA-4 and TIM-3 to negatively regulate the immune system (reviewed by [37]).

## 3. Adoptive Cellular Immunotherapy—CAR-T Cell Therapy

Adoptive cellular immunotherapy aims to exploit cells of the patient’s own immune system to attack cancer cells, via the specific recognition of surface-expressed antigens. This approach has now become a standard anti-cancer treatment [2] in its newest form: Chimeric Antigen Receptor (CAR)-T cell therapy. In this approach, autologous T cells are engineered ex vivo to express CAR molecules, which are non-MHC-restricted antigen receptors. CAR molecules are generally composed of the single-chain variable fragment (scFv) of an antibody fused via a transmembrane domain to T-cell intracellular signalling domains. These signalling domains include TCR-derived molecules such as the CD3 zeta signalling chain, together with costimulatory signals such as CD28, 41-BB or OX40 [38]. Expanded populations of these gene-modified T cells are then infused back into the patient [39].

The starting materials for CAR-T cell manufacture are peripheral blood mononuclear cells (PBMCs) isolated from whole blood or from leukapheresis products. T cells are selected and activated in vitro through the CD3 and CD28 molecules, ready for transduction with a viral vector or electroporation with DNA or mRNA encoding the CAR molecule [40,41,42]. Among these gene transfer methods, lentiviral vectors are currently the most commonly used for transducing T cells [43]. After transduction, CAR-T cell products are cultured in medium generally supplemented with IL-7 and IL-15, or IL-2 only, allowed to expand for 10–14 days and cryopreserved for later therapeutic use [44,45]. In most clinical studies, the times allowed for cell expansion are similar and depend on the strong proliferation of the starting cell populations in order to generate a sufficient number of cells for therapeutic doses [44,46]. We discuss the importance of the starting cell population later in this review. CAR-T cells are infused into patients at pre-defined doses and are designed to recognise tumour-associated antigens and induce cancer cell death [6].

The clinical application of CAR-T-cell therapy in haematological cancers has shown impressive results, with durable remission rates of 30–40% in patients with heavily pre-treated disease [6,47]. This field is moving rapidly, with early reports of effective therapy against acute and chronic lymphoid leukemias in 2011 [48,49,50] and the first clinical trial outcomes reported in 2014, when Maude et al. reported fully on a new therapeutic strategy for r/r B-cell acute lymphoblastic leukaemia (B-ALL) using anti-CD19 CAR-T cells [51]. In 2017, this approach was translated widely into clinical practice for both r/r B-ALL and diffuse large B-cell lymphoma (DLBCL) with US FDA approval of the two CAR-T cell therapies, tisagenlecleucel and axicabtagene ciloleucel [6,47].

The efficacy of clinical CAR-T cell therapies depends upon efficient expansion and persistence of the transfused T cells. Therefore, reinvigorating exhausted CAR-T cells post-infusion is one currently discussed strategy to further increase treatment success. Several studies have shown that sequential CAR-T cell and PD-1 blockade therapy may re-activate the expansion of CAR-T cells in a subset of patients exhibiting poor persistence of transferred cells [52,53]. CAR-T cell re-expansion peaks were observed in peripheral blood, and infused CAR-T cells exhibited prolonged persistence (twice as long as the first CAR-T cell infusion). Several multicentre trials are now opening to investigate the optimal timing for infusion of immune checkpoint inhibitors in lymphoma patients who have relapsed or progressed following CAR T cell therapy (ClinicalTrials.gov identifiers: NCT03310619 and NCT03630159).

However, despite the convincing clinical activity of CD19-specific CAR-T-cell therapy, the adoptive transfer of engineered T cells has not been as successful for the treatment of solid cancers [54,55] and the addition of PD-1 inhibition did not further enhance expansion or persistence of GD2-specific CAR-T cells in patients with advanced neuroblastoma [56]. Therefore, optimising the design and manufacturing of CAR-T cell therapy for solid cancer patients is an area of intense investigation. Although many factors must be considered in developing a new CAR-T therapy, including identification of appropriate target antigens and design of ideal CAR structures, here we consider one aspect that is often overlooked: the status of the patient’s peripheral immune system, and the way that it can be affected by chemotherapy treatments.

## 4. Classes of Cytotoxic Chemotherapy Drugs

Cytotoxic chemotherapy drugs can damage RNA and DNA molecules and target processes of cell division of both normal and cancer cells. However, poor DNA repair and/or higher growth fractions or rates of cell proliferation make cancer cells more susceptible to the cell killing and growth-inhibitory effects of cytotoxic drugs. Different classes of cytotoxic drugs depend on distinct mechanisms of action to exert these effects. These drug classes include alkylating agents, platinating agents, antimetabolites, mitotic inhibitors and anti-tumour antibiotics [57].

Alkylating agents such as cyclophosphamide and temozolomide prevent cells from dividing by causing DNA damage, which may include breakages, cross-linking of DNA strands and abnormal DNA base pairings, thus affecting cells in all phases of the cell cycle. Platinums, such as cisplatin, oxaliplatin and carboplatin, represent another important class of non-cell cycle-selective anti-cancer drugs. On the other hand, antimetabolites mimic substances that cells need for dividing or growing, and these drugs can only destroy cancer cells when the cell’s DNA is being replicated. Examples of antimetabolites are purine antagonists (6-mercaptopurine), pyrimidine antagonists (5-fluorouracil, gemcitabine, capecitabine) and folate antagonists (methotrexate and pemetrexed). Mitotic inhibitors are often derived from plants and act as anti-microtubule agents or topoisomerase inhibitors, thus blocking cell division. Paclitaxel, docetaxel, vinblastine, etoposide and irinotecan are examples of chemotherapy drugs based on plant alkaloids. Anti-tumour antibiotics bind to DNA and prevent RNA transcription, thereby blocking protein synthesis, which is critical for cell survival. Doxorubicin, epirubicin, daunorubicin and bleomycin are examples of anti-tumour antibiotics. Similar to antimetabolites, the anti-cancer effectiveness of mitotic inhibitors and anti-tumour antibiotics is cell-cycle-phase-specific.

Rather than the use of single agents or even the sequential use of such agents, combination chemotherapy regimens have been one of the most important advances in cancer treatment. These approaches exploit non-overlapping mechanisms of action (MOA) and non-overlapping toxicities to delay or overcome therapeutic resistance and thus improve tumour control rates and patients’ tolerance of treatment [58].

## 5. Effects of Chemotherapy on Circulating Immune Cells

Cells of the immune system can be affected by cytotoxic drugs in many ways. The numbers and intrinsic quality of blood-borne immune cells are critical for the effectiveness of subsequent cell-based immunotherapies. This section therefore considers the effects of commonly used chemotherapy drugs on different circulating immune cell populations.

Most chemotherapy agents produce well-recognised and immediate cytotoxic effects on haematopoetic precursors in bone marrow, resulting in subsequent short-term neutropenia. However, less well-known is the fact that many cytotoxic drugs also induce longer-term effects on certain immune populations that occur after leukocyte numbers have rebounded. Here, we will discuss both short- and long-term effects of chemotherapy agents on different leukocyte populations. More specifically, because we are interested in understanding how prior chemotherapy could affect the subsequent production and function of CAR-T cells, we focus specifically on the circulating leukocyte pool, rather than cells in the tumour microenvironment. We have also restricted our discussion to chemotherapy agents administered for their anti-cancer effects and excluded the specific example of lymphodepleting cyclophosphamide and fludarabine chemotherapy, which is the standard preparative regimen preceding CAR-T cell infusions in patients with r/r B-cell malignancies.

### 5.1. Chemotherapy Enhances the Activity of Circulating CD8^+^ T Lymphocytes and Promotes a Regeneration of the Effector Memory Population

Numerous papers have assessed the immediate (during the course of treatment) and long-term (treatment completed) effects of chemotherapy on the number and function of T cells, because this is important for responses against pathogens and cancers as well as subsequent immunotherapy using immune checkpoint inhibitors, adoptive transferred T cells or cancer vaccines [59,60,61,62,63]. In this chapter, we will discuss the overall survival and recovery of CD8^+^ T cells after exposure to various chemotherapy drugs, both in clinical trials and ex vivo. Cytotoxic chemotherapy to cure and palliate cancer has developed from single agents to combinations of drugs with different MOA, from conventional dose to increased dose intensity, or even low-dose regimens, with the ultimate aims of improving tumour control while minimizing adverse effects, especially on bone marrow, the gut and the immune system, so that effective treatment can continue without modification of dose and schedule.

Cytotoxic drugs at different doses can affect circulating CD8^+^ T cells differently. Mackall et al. reported that high-dose cyclophosphamide resulted in an immediate decrease by half of pre-treated peripheral CD8^+^ T cell numbers during the first cycle of therapy and that CD8^+^ T cell numbers did not recover after three cycles in patients with brain tumour and non-Hodgkin lymphoma, or after 10 cycles in patients with sarcoma [64]. Similarly, patients with malignant mesothelioma and advanced non-small cell lung cancer (NSCLC) showed a rapid reduction in CD8^+^ T cell counts after the first week of receiving platinum-based agents coupled with paclitaxel or gemcitabine, and then the number of CD8^+^ lymphocytes continued to decline at a slower rate throughout the later three cycles [62]. In contrast, Scurr et al. reported that the mean absolute number of CD8^+^ T lymphocytes is elevated in the first week of therapy, then dropped back and was maintained at around baseline levels during the first course of treatment with low-dose cyclophosphamide in metastatic colorectal cancer patients [9]. Interestingly, several studies have revealed that, although temporarily decreased during the first cycle of chemotherapy, the numbers of CD8^+^ T cells return to baseline over time. This was observed in patients receiving cisplatin-based chemotherapy with low-dose cyclophosphamide treatment [65,66] and in a cohort of patients receiving either a high dose of cyclophosphamide or epirubicin plus paclitaxel [65,66]. For example, in the study by Mackall et al., the number of CD8^+^ T lymphocytes dropped in blood in the first two weeks but rapidly recovered one month after intensive chemotherapeutic regimens. Within three months post-therapy, total CD8^+^ T cell numbers returned to baseline and then stabilised, with a median doubling time of 12.6 days [65].

Importantly, most studies consistently describe a rapid proliferation of CD8^+^ T cells after one cycle of chemotherapy, although the cell pool may not fully recover [61,62,66]. Analysing subpopulations of CD8^+^ T cells that accounted for the rapid expansion, researchers reported that effector cells (CD8^+^ CD28^−^ or CD8^+^ CD62L^−^) exponentially regenerated after two weeks following treatment with doxorubicin, carboplatin, paclitaxel, and cyclophosphamide [65,66,67]. These cell populations significantly expanded during the course of chemotherapy, and cell numbers at the third cycle were higher than at diagnosis [65,66,67]. Fadul et al. subclassified effector T cells as effector memory (EM) CD8^+^ T cells, which are negative for both CD45RA and CCR7 markers, and terminal effector (EMRA) CD8^+^ T cells, which also lack CCR7 but have regained expression of CD45RA [7]. They found that EM CD8^+^ T cells were significantly increased, whereas EMRA CD8^+^ T cells decreased markedly in a cohort of glioblastoma patients after completion of external beam radiotherapy and temozolomide chemotherapy. Wu et al. also showed that the proportion of memory CD45RO^+^ CD8^+^ T cells is maintained in patients’ peripheral blood in the first two weeks of treatment with paclitaxel and carboplatin, then significantly enhanced in the following week before returning to baseline at the end of cycle 1 [68]. Interestingly, most studies have reported both the proportion and the absolute number of naïve (CD45RA^+^ CCR7^+^) and central memory (CM) (CD45RA^−^ CCR7^+^) CD8^+^ T cells neither decrease nor increase four weeks after initiating alkylating agent-based or anti-microtubule-based chemotherapy [7,65,66,67]. These data suggest that effector memory CD8^+^ T cells are the main source for the re-expansion of the CD8^+^ T-cell population after chemotherapy.

Corresponding to these findings, Turtle et al. identified a mechanism underlying the strong resistance of memory CD8^+^ T cells to cytotoxic treatment [69]. A distinct proportion of CM and EM CD8^+^ T cells over-express ATP-binding cassette (ABC)-superfamily multidrug efflux proteins, which protect cells from toxic agents, and these subpopulations were strongly resistant to daunorubicin-induced apoptosis [69]. Importantly, CD8^+^ memory T cells with high efflux capacity were shown to proliferate strongly with IL-7/IL-15 supplements in vitro and in patients with chemotherapy-induced lymphopenia [69]. This again supports the contribution of these cells to re-forming the T cell memory subpopulation after chemotherapy [69,70].

Interestingly, chemotherapy agents may also induce apparent anti-cancer T cell responses. In a phase I/II clinical trial in metastatic colorectal cancer patients, which measured the effects of low-dose cyclophosphamide on T-cell subsets, a remarkable increase in the percentage of T cells producing perforin/granzyme B and IFN-γ was shown on day 4 of treatment, and the elevated frequencies of these populations were maintained for the first month [9]. Similar results were also reported in a study on metastatic ovarian cancer patients who were treated with dose-dense weekly paclitaxel in combination with carboplatin [67,68,71]. Interestingly, Bellone et al. found that both spontaneous and induced IFN-γ release by CD8^+^ T cells after each course of combined chemotherapy with cisplatin, gemcitabine and 5-fluorouracil (up to four cycles) are unchanged in comparison with pre-treatment values, indicating a preservation of effector function throughout the treatment course [72].

Together, these data suggest that CD8^+^ T cells are relatively resilient to the effects of many cytotoxic chemotherapy agents. Although these agents may induce a transient decrease in blood CD8^+^ T cell numbers, the population recovers well in most cases, largely due to an expansion of the effector memory pool. Moreover, some drugs even induce a transient increase in CD8^+^ T cell number. Furthermore, antineoplastic drugs may enhance the cytotoxic activity of CD8^+^ T cells, which may contribute to their anti-cancer activity. We therefore conclude that, following a recovery period, the CD8^+^ T cell pool may actually be strengthened following certain chemotherapy regimens, due to increased cytotoxic activity and a population shift from terminal effector cells (EMRA) to effector memory (EM) cells.

### 5.2. Chemotherapy Produces Inconsistent Changes among Blood CD4^+^ T Lymphocytes

In contrast to the relatively consistent patterns observed for CD8^+^ T cells, the effects of various chemotherapeutic agents on CD4^+^ T cells varied substantially among studies. Blood samples collected from glioblastoma patients four weeks after completing temozolomide treatment, or from breast cancer patients up to nine months after receiving docetaxel, doxorubicin and cyclophosphamide, had an approximate 50% reduction in the number of CD4^+^ T cells compared to pre-treatment levels [7,73]. Similarly, Hakim et al. showed a prolonged reduction in CD4^+^ T cell number in another cohort of breast cancer patients up to 12 months after high dose of FLAC (5-Fluorouracil, leucovorin, doxorubicin and cyclophosphamide) chemotherapy. There was a particularly severe depletion of CD4^+^ T cells in patients who had docetaxel in the treatment plan, and these cell numbers did not recover after cessation of chemotherapy [73]. In contrast, the percentage of peripheral CD4^+^ T cells was shown to steadily increase every week during a six-week treatment period in pancreatic carcinoma patients who received a combination of cisplatin, gemcitabine and fluorouracil [72]. Another cohort of gastric cancer patients treated with the FLEEOX regimen of five different cytotoxic drugs had no change in the proportion of CD4^+^ T cells after two cycles of therapy [74]. Further analysing CD4^+^ subsets, Fagnoni et al. and Verma et al. found that cyclophosphamide and paclitaxel caused a significant drop in both the mean absolute number and the percentage of naïve CD4^+^ T cells, defined as CD45RA^+^ CD45RO^−^ CD62L^+^ [66,73]. Similarly, Hakim et al. revealed a loss of greater than 90% of CD45RA^+^ CD4^+^ cells after the first cycle of therapy [75]. This subset gradually increased but was still not fully replenished up to nine months post-chemotherapy [75]. Conversely, memory CD4^+^ T cells (CD45RO^+^) showed only a low-level depletion during chemotherapy and then progressively increased to more than 80% of pre-treatment levels in the study of Hakim et al., while in the study of Verma et al., these cells even increased by 20% in their proportion during nine months of follow up [66,73]. These findings show unexpected and inconsistent changes in CD4^+^ T-cell populations post-chemotherapy, probably because of the different types of chemotherapy drugs used and the differences between patients and their respective types of cancers.

Noticeably, CD4^+^ and CD8^+^ T cells use different pathways to regenerate, which may contribute to the differences seen in response to chemotherapy. The thymus plays a major role in the regeneration of CD4^+^ T cells but not of CD8^+^ T cells [65,75]. The thymus undergoes atrophy with age and is disrupted by chemotherapy drugs, which could explain the observed deficiencies in CD4^+^ T cells, especially the CD4^+^ CD45RA^+^ subset, in most studies involving aged patients [76,77]. Conversely, CD8^+^ T cells demonstrate thymic-independent restoration, with high proliferative capacity from the CD8^+^ CD28^−^ repertoire [65]. These findings should be carefully noted because the pool of regenerated CD8^+^ T cells could determine the effectiveness of engineered cytotoxic T cells.

### 5.3. Chemotherapy Has Diverse Effects on Circulating Natural Killer Cells

Natural killer (NK) cells are innate effector cells that play an important role in controlling the development of neoplasia [78]. Different chemotherapy regimens have diverse quantitative and qualitative effects on NK cells [73,79,80]. In testicular cancer patients, the mean absolute number of NK cells (defined as CD16^+^/CD56^+^) was significantly decreased after initiating treatment with a combination of cisplatin, bleomycin, etoposide and granulocyte-macrophage colony-stimulating factor (GM-CSF), then slowly increased but did not return to pre-treatment levels by the end of the first cycle (day 21) [80]. In contrast, in breast cancer patients, the combination of doxorubicin with fluorouracil and cyclophosphamide was reported to result in a significant increase in NK cell number after the first two cycles, with a continued increase until the end of treatment (at cycle 6) [79]. In another study of breast cancer patients who received a similar chemotherapeutic regimen, the mean absolute numbers of NK cells after ceasing treatment was notably lower than pre-treatment levels, but these numbers recovered completely during the following six months [73]. Interestingly, antimetabolite chemotherapy drugs in these studies were reported to support and boost the capacity of NK cells to kill target cells, while cisplatin and alkaloid plant-derived drugs such as paclitaxel, docetaxel and vinblastine inhibited NK cell-mediated killing in vitro [80,81]. In short, chemotherapy drugs can cause transient changes in NK cell number, but this population can, in many cases, be restored to physiological cell numbers after finishing treatment. In addition, both positive and negative effects of different types of chemotherapy drugs on NK cell function have been described.

### 5.4. Chemotherapy Depletes Circulating Regulatory T Cells

Regulatory T cells (Tregs; characterised as FoxP3^+^ CD25^+^ CD4^+^) are potent suppressive cells in the immune system and considered to constrain anti-tumour immune responses [82]. Interestingly, chemotherapy can induce changes to this population that may benefit the anti-cancer functions of the immune system [83,84,85]. Two studies in gastric cancer and colorectal cancer patients using the FOLFOX 6 (5-fluorouracil, leucovorin and oxaliplatin) and the FOLFIRI (5-fluorouracil, leucovorin, and irinotecan) regimens identified a significant decrease in the frequency and the number of Tregs among PBMCs seven days after receiving the first dose of treatment, especially in a group of patients who had exhibited a high proportion of Tregs before treatment [83,85]. The reduction in circulating Treg numbers was also seen in NSCLC patients after the first week of treatment with cisplatin-based chemotherapy, and this suppressive effect was maintained until the end of therapy (four cycles) [86]. Similarly, cyclophosphamide was reported to induce apoptosis of Tregs and eliminate the homeostatic maintenance of this population in vitro and in a mouse model [87,88,89]. One study showed that the *FoxP3* and *GITR* genes, which drive the suppressive activities of Tregs, were significantly downregulated in mice treated with cyclophosphamide [88]. In addition, cyclophosphamide-induced depletion of Tregs and loss of suppressive function were shown to rescue immune dysfunction in animal models of cancer [89,90,91]. Thus, chemotherapy significantly reduces regulatory T-cell number and function, which leads to reversal of immune dysfunction and therefore may facilitate anti-tumour immunity.

### 5.5. Chemotherapy Decreases the Number of Circulating B Lymphocytes

Mature B cells can contribute to anti-cancer immune responses by producing tumour-specific antibodies and by optimising T-cell activation via their presentation of tumour antigens to CD4^+^ T cells [92,93]. On the other hand, a subset of IL-10-producing B cells, called regulatory B cells (Bregs), have also been identified. Bregs can exert immunoregulatory functions, for example, by suppressing IFN-γ production by CD4^+^ T cells in gastric cancer patients [94,95,96]. Both the absolute number and the frequency of all B-cell subsets were significantly diminished over two to 12 weeks in patients undergoing treatment with cyclophosphamide-, epirubicin/doxorubicin-, paclitaxel- and platinum-containing chemotherapy regimens [97,98]. Although absolute cell counts were decreased, studies in patients treated with doxorubicin and cisplatin/carboplatin have shown an enhanced ability of B cells to activate T cells, which is facilitated by the upregulation of the co-stimulatory molecule CD86 on conventional B cells, and the reduced secretion by Bregs of the immunosuppressive factors IL-10 and adenosine [98,99]. Nevertheless, B cells recover in the long term, with cell numbers reaching the pre-treatment level one year after termination of platinum-based chemotherapy [98]. In contrast, treatment with a high dose of methotrexate for six weeks [98,100] or a low dose for six months [98,100] has been shown to result in a decrease in the conventional B cell population, but not Bregs, at the end of treatment. The increased relative proportion of Breg cells and enhanced production of adenosine appeared to protect tumour cells from anti-tumour T cells [98,101]. In short, selected antineoplastic drugs augment functions of multiple B cell subsets, including regulatory B cells, which may counterbalance the remarkable decrease in circulating B cell numbers.

### 5.6. Chemotherapy Increases Blood Monocyte Counts and Preserves Their Functions

CD14^+^ monocytes are bone marrow-derived innate immune cells of the mononuclear phagocyte system. They circulate in the bloodstream, traffic to tissues and can differentiate into tumour-associated macrophages (TAMs) and monocyte-derived dendritic cells (moDCs), which, depending on immune contexture, can play a key role in promoting tumour growth or anti-tumour immune responses [102]. Chemotherapy-induced changes to blood monocytes could lead to changes in the number and function of both moDCs and TAMs, which will in turn affect the cancer microenvironment and immune system function. In one study, the change in CD14^+^ cells in a cohort of metastatic colorectal cancer patients who received FOLFOX, FOLFOXIRI or XELOX (oxaliplatin and capecitabine) regimens as neoadjuvant therapy was evaluated [103]. The total monocyte count was found to be elevated after two treatment cycles. Furthermore, in monocytes isolated after chemotherapy, functions such as TNF-α production, chemokine receptor expression and phagocytic activity did not differ from healthy or pre-treatment populations, indicating that these chemotherapy regimens preserved normal monocyte function [103]. In another study of 104 unresectable colorectal cancer patients who were undergoing palliative chemotherapy, a significant increase in total monocyte counts was observed eight weeks after commencing chemotherapy [104]. All patients were treated with a combination of oxaliplatin, irinotecan, 5-FU and leucovorin [104]. Thus, clinical measurements consistently demonstrate that chemotherapy increases monocyte frequency and preserves monocyte activity during treatment. However, these studies did not include data for long-term changes in monocyte counts after termination of chemotherapy.

### 5.7. Chemotherapy Reduces Circulating Myeloid-Derived Suppressor Cells

Myeloid-derived suppressor cells (MDSCs) are immature myeloid cells that are arrested at various stages during differentiation [105]. MDSCs are classified into granulocytic and monocytic MDSCs according to their originating lineages [105]. MDSCs are known to exert immunosuppressive effects in cancer patients, and accumulation of these populations in blood or the tumour microenvironment correlates with the extent of clinical cancer stage and a poorer prognosis [106,107]. Interestingly, chemotherapy has been shown to inhibit MDSC populations in mouse models and in cancer patients. The absolute number and frequency of these suppressive cells was significantly reduced after six cycles of treatment with gemcitabine and fluorouracil and tended to decrease after three cycles of a platinum-based regimen [106,108,109,110]. However, all post-treatment blood samples were obtained approximately 14 days after the last dose of chemotherapy, which suggests that the suppressive effects of agents on MDSCs may not be durable. Indeed, these studies were limited by relatively small sample sizes, and subsequent randomised phase II and phase III clinical trials examining the effects of gemcitabine and capecitabine on the immune system revealed that these two drugs did not suppress levels of MDSCs after six cycles of treatment [84,111]. Nevertheless, the combination of gemcitabine and GM-CSF blockade can inhibit ex vivo the function of MDSCs derived from pancreatic cancer patients [112]. Therefore, chemotherapy agents have the potential to reduce MDSC-induced immune suppression, but further research is required to better understand the durability of these effects.

### 5.8. Summary

The differential effects of cytotoxic chemotherapeutic agents on individual immune cell subsets may either help or hinder anti-tumour immune responses, as summarised in Table 1. Overall, following a period of recovery and repopulation, the net effect of cytotoxic drugs on circulating immune cells is likely to be positive. This is due to an increase in the effector function of CD8^+^ T cells and a shift toward an effector memory phenotype, together with reduced number and function of immunosuppressive Treg and MDSC. However, the effects of chemotherapy agents on B cells, NK cells and effector CD4^+^ T cells are less consistent, and may arguably have pro-tumour or anti-tumour outcomes.

These clinical observational studies have not incorporated investigations of cell cycle-specific effects of particular cytotoxic drugs on, for example, the repopulating ability among individual immune and myeloid cell subsets. Furthermore, it must be acknowledged that the generalised conclusions made here will not necessarily apply to every agent, and will also be dose-dependent. An improved understanding of the effects of conventional chemotherapy on quantitative and qualitative aspects of immune system function and anti-tumour immunity may be obtained by further prospective and controlled clinical studies using well-defined chemotherapeutic regimens.

## 6. Potential Supra-Additive Anti-Tumour Effects of Chemotherapy and Adoptive T-Cell Therapy

Although long considered a potent anti-cancer treatment, a common perception of conventional chemotherapy is that its potentially severe side effects include a substantial impairment of immune function [75,113,114]. To the contrary, in this review, we have described significant immunomodulatory effects of cytotoxic agents that may help to revive a cancer patient’s systemic immune system before CAR-T cell therapy. We will discuss the practical implications of these observations in the next section. However, it is also worth highlighting the potential for CAR-T cells and chemotherapeutics to act synergistically by administering both types of therapy within the same timeframe [115,116]. This is because, in addition to the systemic effects discussed so far, chemotherapy drugs can also have diverse effects on the tumour-immune microenvironment, which can, in turn, affect local CAR-T cell activity. These effects have been reviewed in detail by others [117,118,119], but two key aspects are worth highlighting here. First, chemotherapy drugs can induce immunogenic cell death (ICD) of cancer cells, a phenomenon in which molecular responses of invariant patterns accompany tumour cell death and prompt innate immune responses [120,121,122]. Second, chemotherapy agents can significantly affect the expression of key immune checkpoint molecules such as PD-L1 [122], which in turn can affect CAR-T cell function.

The potential value of combination chemotherapy/T-cell therapy approaches is also supported by preclinical data showing that exposure of cancer cells to cytotoxic drugs in vitro can result in enhanced susceptibility to cytolytic mediators used by immune cells [123]. At sub-cytotoxic doses, chemotherapeutic drugs increase the permeability of cancer cell membranes to granzyme B and induce a several-fold upregulation of mannose-6-phosphase receptors (MPR), which are receptors for granzyme B on target cells [123,124]. As a result, drug-treated cancer cells show an increase in granzyme B uptake and apoptosis after incubation with specific cytotoxic T lymphocytes (CTLs) [123]. Similar results were also observed when using non-specific CTLs, suggesting that these chemotherapy-induced effects permit CTLs to bypass the requirement of specific antigen recognition to destroy cancer cells [123].

Considering these in vitro findings, it is not surprising that synergistic effects of chemotherapy and T-cell therapy have been observed in animal models. For example, Alizadeh et al. administered doxorubicin and T-cell therapy in a mouse model of breast cancer [125]. Non-specific Th1 or Th17 cells were generated from naïve spleen cells in vitro and infused into mice two days after they were treated with doxorubicin. The combination therapy significantly impaired tumour growth and minimised the number of metastatic nodules compared to either doxorubicin or T-cell therapy alone. Importantly, chemotherapy followed by adoptive cell therapy regimens could induce immunological memory, as indicated by significant increases in memory CD4^+^ and CD8^+^ T cells in cured mice and the ability to reject the same tumour cells upon re-challenge [125,126]. Thus far, however, clinical data reporting the combination of conventional chemotherapy and CAR-T therapy for solid tumours are lacking. Indeed, recent publications have attempted to use mathematical modelling to determine how to maximise the positive immunomodulatory effects of chemotherapy [127,128]. Further clinical studies testing combinations of chemotherapy with the newest and most successful form of T-cell therapy, CAR-T cells, are clearly warranted.

## 7. Prospects for Manufacturing CAR-T Cells from Solid Cancer Patients after Chemotherapy

The US FDA-approved CAR-T cell therapies, tisagenlecleucel and axicabtagene ciloleucel, have achieved remarkable results, mainly in heavily pre-treated children and young adults with r/r B-ALL and DLBCL [129,130]. These CAR-T cell products have usually been manufactured successfully after the failure of standard or salvage treatments [129], thus indicating that prior chemotherapy has not precluded the generation of clinically effective CAR-T cell products. In an earlier report, Kochenderfer had shown that PBMCs from chronic lymphoblastic leukaemia (CLL) patients who underwent chemotherapy can generate highly functional CAR-T cells in vitro, with a high proliferation rate and transduction efficiency [131]. The time between the last treatment of chemotherapy drugs and PBMC collection for CAR-T cell manufacture is not clear from this report, but some patients did show lymphopenia before blood collection [131].

Together, these observations suggest that satisfactory yields of functional CAR-T cells can be generated from patients with haematological malignancies even when post-chemotherapy PBMCs constitute the starting population. It remains an open question whether this observation also holds true for other cancer patients treated with different chemotherapy regimens. However, recent listings of recruiting clinical trials (clinicaltrials.gov identifiers: NCT01454596, NCT01583686, NCT03799913) of CAR-T cell therapy in patients who have recurrent or progressing brain tumours, mesothelin-expressing tumours (metastatic pancreatic and other types of cancer), and ovarian cancers, and who have been recruited after at least first-line standard chemotherapy, suggest that manufacture of the CAR-T cell products is feasible in these patient populations. Indeed, preclinical data from the trial NCT01454596 has shown the feasibility of manufacturing functional CAR-T cells with high in vitro cytotoxic activity from these patients [44].

Interesting clinical studies of the metabolic properties of T cells, which were collected before and after the second cycle of chemotherapy from a wide range of solid cancer patients, indicate that short exposure to chemotherapy did not significantly affect the metabolic profiles of T cells [132]. Similarly, Das et al. suggested that early stimulation of post-chemotherapy T cells in vitro, which is a part of the CAR-T cell manufacturing process, can lessen mitochondrial damage in T cells and may increase the quality of the cell product. To generate potentially effective CAR-T cells, these authors recommended that PBMCs be collected either before or within eight weeks after the first dose of chemotherapy [59].

Thus, manufacturing CAR-T cells from cancer patients treated with a variety of chemotherapy agents appears feasible. In fact, the results discussed earlier in this review (summarised in Table 1) raise the intriguing question of whether CAR-T cells manufactured from blood collected post-chemotherapy may even be superior to those manufactured prior to chemotherapy. There are several time-points along a patient’s treatment journey at which PBMCs could potentially be collected and cryopreserved for subsequent CAR-T cell manufacturing (Figure 1). The findings discussed here suggest that post-chemotherapy banking of PBMCs could result in an optimal CAR-T cell product compared to PBMCs collected before chemotherapy. The exact timing would depend on the agent given and should ensure that the patient has recovered from any short-term lymphodepleting effects. PBMC collection at the time of disease relapse may also be a possibility. However, by this time, some of the positive effects of chemotherapy on blood leukocytes may have been lost or reduced, and the patient may be experiencing substantial systemic immune suppression driven by their cancer burden [133].

## 8. Conclusions

The striking clinical results observed after the use of CAR-T cell therapy in r/r haematological cancer patients continue to encourage the development of CAR-T cell therapy for solid cancer patients. Based on data from a large number of studies and reviews [115,123,125,134], we conclude that certain commonly used chemotherapeutic drugs can favourably modulate the immune system. These agents generally preserve the number of CD8^+^ T cells (after an initial depletion) and in fact can be associated with an increase in their cytotoxic function and a shift toward effector phenotypes. Chemotherapy drugs may also induce a depletion of Tregs and MDSCs in peripheral blood, thereby reducing immunosuppressive activity. In contrast, effects on B cells, NK cells and CD4^+^ effector T cells are more diverse. Thus, the manufacturing of CAR-T cells from blood collected post-chemotherapy may not only be feasible, but perhaps even preferable compared to blood collected before chemotherapy.

Clearly, empirical studies that compare CAR-T cell manufacturing at different times before, during, and after chemotherapy in solid cancer patients will be required to determine the optimal timing both for collection of the starting cell population for CAR-T cell manufacturing, and subsequent infusions of CAR-T cell products, relative to administration of chemotherapy. Relatively simple changes to treatment scheduling could potentially produce beneficial patient outcomes.

## Figures and Tables

**Figure 1 cancers-13-02225-f001:**
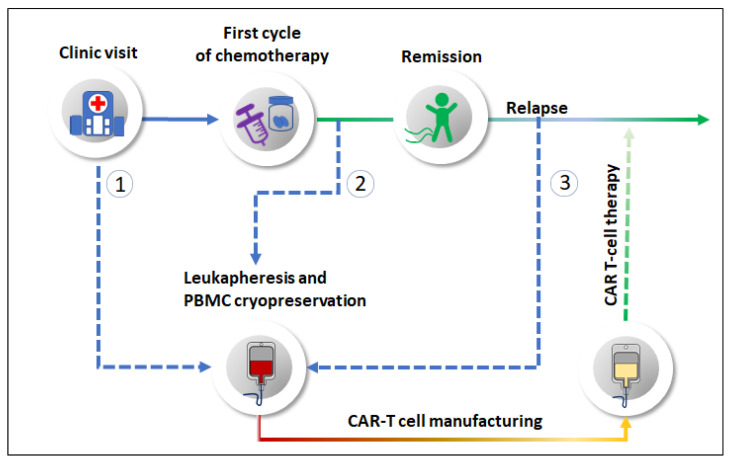
Schematic diagram showing proposed times for making CAR-T cell products in patients treated with cytotoxic chemotherapy. Leukapheresis for CAR-T cell manufacturing could be performed when ① patients are first seen in clinic, ② after receiving the first cycle of chemotherapy or ③ after disease relapse. Based on the information reviewed herein, option ② may be optimal.

**Table 1 cancers-13-02225-t001:** Summary of changes to circulating immune cell populations after exposure to chemotherapy drugs ^1^.

Cell Type	Type of Modulation	Chemotherapy Drugs Used	References
CD8^+^ T cells	Transient decrease (2–4 weeks) in total cell number then recovery toward baseline value (effector memory CD8^+^ T cells contribute the most)	temozolomide	[7]
		cyclophosphamide + paclitaxel/epirubicin + paclitaxel/cisplatin + gemcitabine + 5-FU	[62,65,66,67,68,72]
	Selective survival and expansion of CD8^+^ T cells overexpressing multi-drug efflux proteins	daunorubicin	[69]
	Increase in granzyme B, perforin, and IFN-γ secreting cells; preservation of IFN-γ secretion ability	cyclophosphamide/carboplatin + paclitaxel/cisplatin + gemcitabine +5-FU	[9,68,72]
CD4^+^ T cells	Long term decrease in total cell number (mostly naïve CD4^+^ T cells)	temozolomide	[7]
		docetaxel + doxorubicin + cyclophosphamide	[73]
		5-FU + leucovorin + doxorubicin + cyclophosphamide	[75]
	Increase in cell number compared to baseline at all time points assessed	cisplatin, gemcitabine, 5-FU	[72]
	No change in cell number	5-FU + leucovorin + epirubicin + etoposide + oxaliplatin (FLEEOX)	[74]
Natural killer cells	Decrease in cell number during the first treatment course	cisplatin + bleomycin + etoposide + GM-CSF	[80]
	Increase in cell number at all time points assessed (6 cycles)	5-FU + doxorubicin + cyclophosphamide	[73,79]
	Boost cytotoxic activity	Antimetabolite	[80,81]
Regulatory T cells	Long lasting decrease in cell number	Folinic acid + 5-FU + oxaliplatin	[83]
		FOLFOX/FOLFIRI	[85]
		cisplatin	[84,86]
		docetaxel	[86]
B cells	Decrease in cell number and frequencies of all B cell subsets during chemotherapy, but recovery within one year after treatment ceased	cyclophosphamide, epirubicin, doxorubicin, cisplatin, fluorouracil	[97,98,99]
Monocytes	Increase in cell number during treatment (2 cycles)	FOLFOX/FOLFOXIRI/XELOX	[103,104]
Myeloid-deprived suppressor cells	Decrease in cell number during chemotherapy regimens (up to 6 cycles)	gemcitabine or 5-FU/platinum-based + bevacizumab	[106,108,109,110]

^1^ 5-FU: 5-fluorouracil. GM-CSF: granulocyte-macrophage colony-stimulating factor, FOLFIRI: folinic acid + fluorouracil + irinotecan, FOLFOX: folinic acid + fluorouracil + oxaliplatin, FOLFOXIRI: folinic acid + fluorouracil + oxaliplatin + irinotecan, XELOX: capecitabine + oxaliplatin.

## Data Availability

Not applicable.

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
