# Peer review of "Effects of Chemotherapy Agents on Circulating Leukocyte Populations: Potential Implications for the Success of CAR-T Cell Therapies"

_cancers, 2021, doi:10.3390/cancers13092225_

Round 1

Reviewer 1 Report

This is another review of many on a hot and highly competitive topic, CAR T cell therapy. By reviewing and discussing the effects of conventional cytotoxic agents on immune cell populations it addresses a novel and highly relevant aspect. The question how to combine CAR T cell therapy with the alternative treatment modalities to maximize their potential is a burning issue in the field, and profound insights into this issue could substantially contribute to the translational advancement of CAR T cell therapies.

After the well-written introduction, the reader expects to gain practical knowledge and informed suggestions how to best combine CAR T cell therapy with individual cytotoxic agents and their combinations by optimal order and timing.

But instead of working out the distinct effects of individual agents on cell types most relevant for the response to CAR T cells, the review cites numerous examples for the effects of various regimens on a wide variety of mostly bloodborne cell types, even ending with neutrophil granulocytes very obviously affected by the effects of antiproliferative cytotoxic chemotherapy on the hematopoietic compartment. Overall the article lacks focus, and the chosen examples often seem arbitrary. Chemotherapy is used in an oversimplified manner as an umbrella term, not acknowledging the variability of high and low intensity regimens and the diverse effects of individual agents and their individual combinations that substantially complicate general conclusions.

Further major remarks:

Page 2, chapter 2. "Immune evasion of cancer": Besides loss of (preexisting) immunogenicity, some tumors have a low burden of somatic mutations, e.g. many sarcomas. They can evade T cell recognition simply by a lack or paucity of tumor neoantigens. This aspect is missing.

Page 3, chapter 3.:

At least briefly, the concept of CAR-mediated T cell activation by combined TCR-derived and costimulatory signals should be introduced. Since the aspect of the various CAR T cell designs is beyond the focus of the manuscript, the authors should refer to alternative literature that provides background information. Ref 6 does not appear to be the correct choice here.

The first clinical outcomes were reported in 2011, by Porter et al. and Kalos et al. and by Grupp in 2011, all in the NEJM. And the first phase 1 dose-escalation trial was published by Lee et al. in Lancet 2014. I agree that Maude et al. 2014 is a key paper, but I suggest to reword this to adequately acknowledge the early and concomitant reports.

Page 3, chapter 4.: The principle of chemotherapy could be more elegantly introduced, by stating the mechanism of action these agents have in common, targeting cell division, before introducing the individual classes. Moreover, the concept of combination therapy of various agents of different classes should be introduced. 

Page 4, chapter 5.:

This chapter describes effects of anti-proliferative chemotherapy on circulating immune cell populations (with the exception of MDSCs which appear to be out of place in this chapter), while the subsequent chapter 6 focuses on the tumor microenvironment. It is not clear to the reader that a paragraph on the TME will follow. Since many of the immune cell populations are found both in the periphery and in the TME, where they may be differently affected, the structure is not logical and should be reconsidered.

Page 5, chapter 5.1.:

This paragraph appears biased by citing a few examples in which CD8+ T cells rapidly recover after individual chemotherapy regimens.  Other reports of substantial T cell immunodeficiency after cytoreductive chemotherapy exist, e.g. reviewed by Mackall C in Stem Cells, 2000, and more recently by Velardi et al., Nat Rev Immunol 2020. Especially the introductory statement that "the number and function of CD8 + T cells are well preserved after exposure to various chemotherapy drugs" should be reworded, since even in most of the examples a temporary compromise of T cell immunity was found, and recovery may not reestablish the quality of the pre-chemotherapy repertoire and function. The title with its strong and controversial statement is not adequate. Since the CD8+ T cell population is especially relevant in the context of the manuscript, this paragraph should be as comprehensive as possible, providing a higher number of examples, maybe with help by a revised Table 1.

I also suggest to introduce the concept of thymic and extrathymic T cell expansion earlier in the manuscript and point out the differences for CD8 and CD4 T cells. Even though CD8 cells preferentially rely on extrathymic (and CD4 cells on thymic) expansion for recovery, in my understanding this is not exclusive.

Page 7, chapter 5.6.:

Potentially more relevant in solid tumors than circulating monocytes are tumor-associated macrophages (TAMs). Experimental reports exist how these can be affected by individual chemotherapeutic agents, with large differences among agents. Even in the subsequent paragraph focusing on the TME (chapter 6.), TAMs are only briefly mentioned. And it is unclear why MDSCs are listed in chapter 5 whereas other resident immune cell populations are missing.

Page 8, 5.9.

The conclusion that “the net effect of cytotoxic drugs on the adaptive immune system is likely to be positive” is not adequately justified, especially since the vast majority of information to this point regards circulating blood, not the tumor microenvironment with its higher relevance for antitumor immune responses.

The potential consequences of the exemplified effects of “chemotherapy” on the individual circulating immune cell populations on CAR T cell therapy, as indicated in the introduction, are not discussed.

Table 1: The contents of the text and the table are largely redundant. The authors should consider collecting even more comprehensive data to list in the table, then using the text to point out the key features emerging from the individual examples.

Page 11, chapter 7: While the authors state that they will exclude the effects of standard so-called lymphodepleting chemotherapy on adoptive T cell transfer, they have mixed in an example (Chamoto et al.) that refers to preparative cyclophosphamide.

The Yamada et al. example also is not well chosen since the comparison with historic control illustrates (at best) the additional effect of TCR-gene modified T cell transfer compared to a standard chemotherapeutic regimen, not vice versa.

Chapter 8: The effects of individual cytotoxic agents on the starting material for CAR T cell manufacturing very likely are diverse, and data regarding the use of individual agents with known T cell suppressive function, e.g. methotrexate, prior to leukapheresis do not exist. Even for hematological malignancies, time intervals between last dose of chemo and T cell collection are strongly recommended. Experimental data could be very helpful to understand how a patient could be optimally prepared for the collection of highest quality T cells for CART manufacturing, but these data are still missing. Unfortunately, neither the conclusions nor Figure 1, with a solid tumor patient apparently receiving a CART product manufactured at a non-defined time-point after chemotherapy while in remission, provide substantial additional knowledge.

Reviewer 2 Report

Truong et al nicely reviewed the effect of different chemotherapeutic agents on the tumor microenvironment and different white blood cell types. The authors concluded that successful CAR T cells manufacturing is potentially feasible after receiving chemotherapy for patients with solid tumors.

Major comments:

1- At this time, Kymriah is FDA approved for the treatment of CD19 positive B ALL (in children and young adults) and adults with B lymphomas. Yescarta is approved for B lymphomas in adults. Both products have a very clear washout time for specific categories of chemo/immunotherapeutic agents prior to leukapheresis as this will impact the successful manufacturing of CAR T cells. For example, pegylated asparaginase cannot be given within 4 weeks prior to leukapheresis. Also, cyclophosphamide (non lymphodepleting), vincristine and methotrexate (at a defined dose) has to be discontinued at least 14 days prior to leukapheresis. There are other agents that need to be stopped at certain time points prior to the leukapheresis for Kymriah and Yescarta. If you concluded that chemotherapies (at least most of them) with not affect CD8 quality much then how do you explain the required washout time for the CAR manufacturing of kymriah and Yescarta? Can you please provide more data if available to address this question? The title and abstract are both a little confusing as the reader may think you are giving both CAR T and chemo/immuno at the same time which by the way may be worth mentioning as for example patients with CD19 CAR T cell exhaustion post infusion may benefit from check point inhibitors (Pembrolizumab to augment response to CD19-targeted chimeric antigen receptor (CAR) T cells in relapsed acute lymphoblastic leukemia (ALL), abstract number 187283 ASCO 2017). 

2- Figure 1 s a little confusing. The legend says relapse and the image has the CAR infusion arrow after "remission".  

Reviewer 3 Report

CAR T cell therapy has shown substantial success in hematological malignancies but lacks efficacy in patients with solid tumors. Thus, the quest for methods to improve CAR T cell efficacy in the realm of solid tumors poses a crucial challenge. 

in this manuscript, Truong and colleagues discuss and review the immunomodulatory effects of chemotherapy agents on a variety of immune cells. Moreover, the authors discuss the feasibility of manufacturing CART cells and their theraputic effects in the clinical setting of concurrent conventional chemotherapy.

The is well-structured, provides topical information on the combination of CAR T cell therapy with chemotherapy and is nicely illustrated by a table and a figure.

Minor comments:

1) Table 1: Natural killer cells. Boost cell cytotoxic activity should be changed to boost cytotoxic activity  

2) Page 11 line 44: ...with cytotoxic drugs was associated increases in the number should be changed to with cytotoxic drugs was associated with increases in the number

3) Page 11 line 47 manufacture should be changed to manufacturing

Round 2

Reviewer 1 Report

The extensive review addresses all my Major concerns.

Here are two remaining minor issues:

Lines 233-234: „Most chemotherapy agents induce an immediate depletion of blood leukocytes, which is well recognised and can lead to short-term myelo- and immune-suppression.“ This sentence is not correct. Cytotoxic anticancer agents do not deplete blood leukocytes but hematopoetic precursors in bone marrow, resulting in subsequent neutropenia.

Paragraph 5.8: I suggest to omit the paragraph on neutrophils. The effects of cytotoxic chemotherapy on hematopoiesis is well known and the relevance of neutrophil-to-lymphocyte ratios likely more complex than can be included in this paragraph which, as written, does not contribute valuable information.

Author Response

We are pleased that you find the revised manuscript addresses most of your concerns. Regarding the remaining issues:

  1. Lines 233-234: “Most chemotherapy agents induce an immediate depletion of blood leukocytes, which is well recognised and can lead to short-term myelo- and immune-suppression.” This sentence is not correct …

Response: We agree that this sentence needs some revision, and have corrected it to the following:

Most chemotherapy agents produce well-recognised and immediate cytotoxic effects on haematopoetic precursors in bone marrow, resulting in subsequent short-term neutropenia

  1. Paragraph 5.8: I suggest to omit the paragraph on neutrophils.

Response: We have now deleted this sentence, as well as associated references and the detail about neutrophils in Table 1.

Reviewer 2 Report

Thank you for addressing my comments. 

Author Response

You're welcome. We appreciate your review and feel that the manuscript is much improved after the feedback we received.